# Ecological Factors and Anthropogenic Disturbance May Restructure the Skin Microbiota of Maoershan Hynobiids (*Hynobius maoershanensis*)

**Huiqun Chen** [1,2,3], **Yingying Huang** [1,2,3], **Guangyan Pang** [1,2,3], **Zhenzhen Cui** [1,2,3], **Zhengjun Wu** [1,2,3] **and Huayuan Huang** [1,2,3,*]

1   Key Laboratory of Ecology of Rare and Endangered Species and Environmental Protection, Guangxi Normal University, Ministry of Education, Guilin 541004, China; chq981026@sina.cn (H.C.); 20010523hj@sina.cn (Y.H.); cuizhenzhen910@sina.cn (Z.C.); wu_zhengjun@sina.cn (Z.W.)
2   Guangxi Key Laboratory of Rare and Endangered Animal Ecology, Guangxi Normal University, Guilin 541004, China
3   College of Life Sciences, Guangxi Normal University, Guilin 541004, China
*   Correspondence: huanghuayuan6@sina.cn

**Abstract:** Studies on the skin microbiota of amphibians in different disturbed habitats can clarify the relationship between the skin microbiota composition and environmental factors and have practical implications for the conservation of endangered species. In this study, 16S rRNA high-throughput sequencing was used to profile the skin microbiota of Maoershan hynobiids (*Hynobius maoershanensis*). Our results illustrate that the alpha diversity of the skin microbiota significantly differed among individuals in higher anthropogenic disturbance-degree (HADD) habitats and lower anthropogenic disturbance-degree (LADD) habitats. The diversity of the skin microbiota in forelimb bud-stage tadpoles from HADD habitats was higher than that in their counterparts from LADD habitats. The richness of the skin microbiota in hindlimb bud-stage tadpoles was greater in HADD habitats than in LADD habitats. However, the alpha diversity of the adult skin microbiota did not differ significantly between the two habitats. Furthermore, stepwise regression analysis indicated that the skin microbiota diversity and relative abundance of dominant bacteria decreased with increasing air temperature, water temperature, and pH; conversely, skin microbiota richness increased with increasing humidity. In addition, the relative abundance of dominant bacteria was influenced by anthropogenic disturbance. We conclude that the skin microbiota of Maoershan hynobiids is affected by ecological factors and anthropogenic disturbance, highlighting the importance of the skin microbiota in response to habitat alteration.

**Keywords:** *Hynobius maoershanensis*; cutaneous bacterial communities; amphibia; caudata; 16S amplicon sequencing

## 1. Introduction

The skin microbiota can produce antimicrobial substances or directly stimulate the host immune system to exert immune functions [1,2]. The equilibrium state between the host and skin microbiota commonly facilitates the maintenance of host fitness [3]. Greater microbiota diversity and richness are beneficial to the host, and they can enhance host immunity and direct defenses against invading pathogens [4,5]. Specifically, reduced microbiota stability stemming from habitat environment changes can affect population health [6]. Meanwhile, amphibians adapt to the environment by adjusting the composition and diversity of their skin microbiota [7]. The structure and functions of the skin microbiota are strongly affected by host characteristics, including genetic, anatomic, and physiological characteristics [8]. Based on their amphibious habits, the skin microbiota of amphibians can arise from the soil or water, or possibly from the gut microbiota (feces diffuse in the

environment, which can spread in soil or water to settle on the skin) [9]. The special skin physiology of amphibians keeps them moist, as their skin contains a high number of glands that can secrete mucus [10]. The mucus secreted by their skin glands contains multiple defense molecules (e.g., AMP, alkaloids, lysozyme, and antibodies) [11]. Therefore, the composition of mucus can create different conditions for specific microbiota to colonize the skin [11], and the mucus acts together with microbes to help amphibians resist the invasion of pathogens [12]. Amphibians have their own unique skin microbiota [8]. The bacterial phyla present in amphibian skin include Acidobacteria, Actinobacteria, Bacteroidetes, Firmicutes, and Proteobacteria, which are highly abundant [13–16].

The skin microbiota composition and structure are associated with multiple factors, including ecological factors [17]. For amphibians, ecological factors play an important role in the assembly/maintenance of the skin microbiota [18,19]. Specifically, various environmental conditions and the potential symbiotic microbiota repertoire available in the environment can affect hosts' microbiota communities, consequently leading to differences in host microbiota communities at geographical scales [20,21]. For example, the community diversity of the skin microbiota of the boreal toad (*Anaxyrus boreas*) significantly differs among different populations living in deserts and pine forests [17]. Furthermore, the skin microbiota of *Salamandra salamandra* larvae varies in different habitats, and larvae inhabiting streams have greater skin microbiota phylogenetic diversity [22]. Similarly, their microbiota structures geologically differ, with Burkholderiaceae, Comamonadaceae, and Sphingomonadaceae being enriched in pond larvae [22].

Other ecological factors (e.g., humidity, pH, and temperature) can also affect the skin microbiota structure [23–25]. For example, humidity and the water pH probably affect the production of antimicrobial peptide secretions by the host skin, subsequently influencing the diversity of the skin microbiota [24,25]. Studies have illustrated that the soil pH explained the degree of variation in the compositional abundance of salamander skin microbiota [26]. In addition, the environmental temperature is an important predictor of symbiotic microbial diversity for amphibians because of heterothermy [27,28]. For example, the air temperature can predict the rate of *Batrachochytrium dendrobatidis* (Bd) fungal infection in amphibians. Specifically, the Bd infection rate is decreased at temperatures exceeding 25 °C [29,30]. The skin microbiota of amphibians is more diverse in cold winter conditions and less stable thermal conditions than in warm environments and less variable thermal conditions in winter [31].

The commensal microbiota communities of wildlife are extremely sensitive to habitat fragmentation and pollution caused by human activities [1,32]. Because of the highly permeable skin and strong environmental dependence of amphibians, the composition and structure of the commensal microbiota are susceptible to environmental changes [16,33]. Host environment changes caused by human disturbance can alter the diversity of the environmental microbiota in soil and water, consequently affecting the skin microbiota of amphibian hosts [21,26]. Adverse environmental changes resulting from human activities can cause changes in the symbiotic microbial community that can make the host more susceptible to pathogens [34]. Moreover, dysbiosis of the amphibian skin microbiota increases the sensitivity to pathogen infection and decreases skin defense and immune function [35,36]. For example, because integrated pig–fish farming causes pollution in ponds, the high concentration of fecal coliform bacteria in ponds largely reduces the diversity of the anurans skin microbiota, leading to a high proportion of Bd-facilitative bacteria in the skin microbiota and increasing the prevalence of Bd in the frogs [37]. Similarly, the skin microbial community of *Pelophylax perezi* varies significantly in different environments, and the presence of chemical contamination affects the composition of the skin microbiota [33]. In addition, the antibacterial diversity of *Proceratophrys boiei* skin is higher in fragmented forests than in continuous forests [1].

Maoershan hynobiids (*Hynobius maoershanensis*) (Amphibia, Urodela, Hynobiidae) are exclusively distributed in the Mountain Maoer National Nature Reserve, Guangxi Zhuang Autonomous Region, in China. Maoershan hynobiids have a narrow distribution

area and very few populations in the field (1500–1600 individuals) [38]. This endemic species is listed as a critically endangered species in the IUCN Red List [39]. The habitat of Maoershan hynobiids has been partly developed as an alpine wetland tourist area, leading to substantial human interference [38,40]. Tourism may lead to an adaptive decline or the dysregulation of wildlife symbiotic microorganisms [41,42]. Analyzing the impact of habitats disturbed by human activity on changes in the host skin microbiota is important for understanding how host-associated symbiotic microbial communities respond to changes in adverse environmental conditions [1]. Information on the skin microbiota of animals can both clarify the relationship between the skin microbiota composition and environmental factors [36] and provide a research basis for future provisioned populations. We studied the skin microbiota of Maoershan hynobiids using 16S rRNA high-throughput sequencing. We first described the composition and diversity of the skin microbiota and then investigated differences in the skin microbiota between habitats with higher and lower degrees of anthropogenic disturbance. We used stepwise regression analysis to test the influence of ecological factors (water temperature, air temperature, humidity, water pH, water depth, different habitats) and anthropogenic disturbance on the diversity and richness of the skin microbiota. We examined the influence of these factors and anthropogenic disturbance on the abundance of the dominant phyla and families in the skin microbiota. Finally, the variations in the microbiota of Maoershan hynobiids were discussed by testing the following predictions:

The diversity of skin microbiota varies among different habitats [34], and it is lower in disturbed habitats [43]. We predict that the skin microbiota diversity of Maoershan hynobiids is lower in HADD habitats than in LADD habitats.

External and environmental stress can cause changes in the abundance of skin microbiota [33]. Ecological factors affect the diversity of the amphibian skin microbiota [31]. Therefore, we predict that ecological factors and anthropogenic interference affect the skin microbiota alpha diversity and the relative abundance of the dominant bacteria of Maoershan hynobiids.

## 2. Materials and Methods

### 2.1. Sample Collection and Preservation

The study was conducted in the Mountain Maoer National Nature Reserve, Xing'an County, Guangxi Zhuang Autonomous Region (25°52′ N, 110°24′ E) at an altitude of 1950–2000 m, in China. One of the study sites was a creek ditch within the reserve (1950 m elevation) in a tourist spot and less than 5 m away from the roadway. Another study site was in a still-water pond (2000 m elevation) with over a 200 m distance from the roadway and not a tourist attraction. The skin samples of 24 adult Maoershan hynobiids, 30 forelimb bud tadpoles, and 13 hindlimb bud tadpoles, and ecological factor data from January 2021 to July 2022 were collected from these two sites (Table S1). The Maoershan hynobiids were sampled by a random sampling method. For the skin microbiota sampling, the hands of the researchers and the workbench were sterilized with 75% alcohol, and all procedures were performed with sterile gloves. Each Maoershan hynobiid was washed three times in pure water to remove temporary microorganisms on the skin surface [44]. The whole body of each animal was wiped three times with a sterile cotton swab, and the swabs were placed in sterile preservation tubes. All samples were frozen immediately after collection, transported to the laboratory, and stored at –80 °C [45]. All Maoershan hynobiids were returned to their original locations after sampling.

### 2.2. Anthropogenic Disturbance

Because of the development of tourism, the habitat of Maoershan hynobiids has been partly developed as an alpine wetland tourist area [38]. We defined the anthropogenic disturbance as frequent tourist activities and existing human-made roads near the habitat of Maoershan hynobiids [46]. Similarly, we divided the habitats as having higher or lower degrees of anthropogenic disturbance based on whether they were near tourist

sites and roads. Specifically, the habitats located in tourist attractions and close to roads were classified as habitats with a higher anthropogenic disturbance degree (Figure 1). Meanwhile, habitats not at tourist attractions and away from roads were classified as lower anthropogenic disturbance-degree habitats. We determined the average daily number of tourists in the sampling month as an indicator of the degree of anthropogenic disturbance. Tourist numbers were used to compare the effects of anthropogenic disturbance on the skin microbiota between the two habitats. We obtained the data of tourist numbers from the reserve.

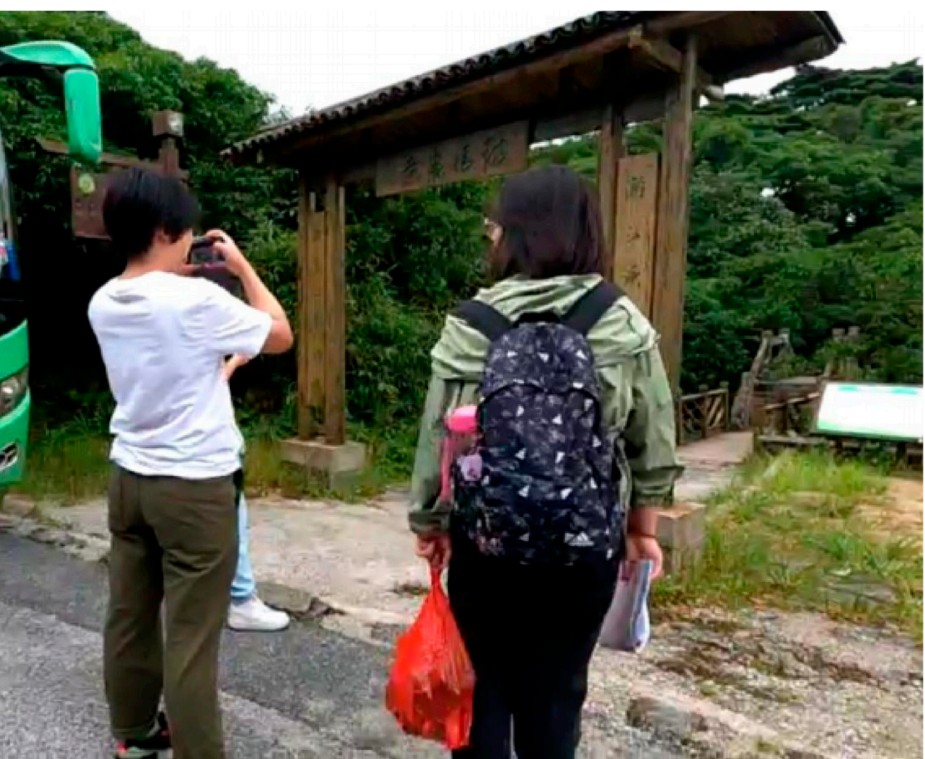

**Figure 1.** Higher anthropogenic disturbance-degree habitat.

### 2.3. DNA Extraction, Amplification, and Sequencing

A FastDNA® Spin Kit for Soil was used to extract the DNA from 67 skin samples. The V3–V4 hypervariable regions of the 16S rRNA gene were amplified by polymerase chain reaction (PCR, GeneAmp 9700, ABI, USA) using the primers 338F (5′-ACTCCTACGGGAGGCAGCAG-3′) and 806R (5′-GGACTACHVGGGTWTCTAAT-3′) [47]. The reaction mixture contained 10 ng template DNA, 4 μL 5×FastPfu buffer, 2 μL 2.5 mM dNTPs, 0.8 μL of each primer (5 μM), and 0.2 μL bovine serum albumin. The initial PCR was performed using Transgen ap221–02 TransStart® FastPfu Fly DNA Polymerase. The PCR products were extracted from 2% agarose gels, purified by AxyPrep DNA Gel Extraction Kit (Axygen Biosciences, Axygen, USA), quantified, and homogenized using QuantiFluor™-ST Blue fluorescence quantification system (Promega) based on the sequencing requirements. The purified PCR fragments were collected and adjusted to an equimolar concentration (sequencing was performed by Majorbio BioPharm Technology Co., Ltd., Shanghai, China), and the paired ends were sequenced (2 × 300) using an Illumina MiSeq platform (Illumina, USA) by the Majorbio Cloud Platform. The datasets presented in this study can be found in online repositories. The names of the repository/repositories and accession number(s) can be found here: https://www.ncbi.nlm.nih.gov/, (accessed on 13 February 2023), PRJNA935520.

*2.4. Data Analysis*

After sample splitting of the PE reads obtained by MiSeq sequencing, the two end reads were first subjected to quality control and filtered according to the sequencing quality, and the overlap relationship between the two end reads was assembled to obtain the optimized data after splicing. The optimized data were noise-reduced using the DADA2 [48] plugin in the QIIME2 (v. 2022.2) process [49]. The noise reduction steps included filtering noise, correcting for sequence errors, removing chimaeras and single sequences, and deduplicating sequences to obtain high-resolution amplicon sequence variants (ASVs) for subsequent analysis. ASV representative sequences were subjected to taxonomic annotation by the SILVA 16S rRNA database (v. 138) using the Bayes species annotation method [50]. To better complete the downstream diversity and composition analysis, each sample was flattened according to the minimum number of sample sequences. The ASVs were classified as chloroplasts and mitochondrial sequences were excluded. In total, 3,214,696 optimized sequences of the hypervariable V3–V4 regions of the 16S rRNA gene were obtained from 67 skin samples, and the average length of the sequences was 423 bp. The sample sequences were leveled in accordance with the minimum number of sample sequences (P1: 16,707). Using a sequence similarity of 100%, 11,352 ASVs were clustered.

Relative abundances of the skin bacterial taxa were expressed as the mean $\pm$ standard deviation, yielding histograms showing phylum-level and family-level community composition. To assess the alpha diversity in the different habitats, we calculated four different metrics (the ACE index, Chao index, Shannon index, and Simpson index) for each group using the Mothur program (v. 1.30.1; https://www.mothur.org/wiki/Download_mothur, accessed on 13 October 2022). Data related to the alpha diversity (including the ACE index, Chao index, Shannon index, and Simpson index of different developmental stages) were tested for normality using the Kolmogorov–Smirnov test. The data during the tadpole stage were normally distributed. Therefore, an independent samples *t*-test was used to compare the differences between HADD habitats and LADD habitats. Conversely, the data of the adult stage were not normally distributed. The Mann–Whitney U-Test was used to compare the differences between the two habitats. For the above test, using SPSS software version 23.0 (SPSS Inc., Chicago, IL, USA), the significance level was set as $\alpha = 0.05$. To improve the linearity, the Shannon, Simpson, ACE, and Chao indices were transformed using $\log_{10}$ (X) [51], and the habitat differences were visualized by R statistical software (R packages "ggpubr", "patchwork", "grid", and "showtext"). Principal coordinate analysis at the ASV level was performed on all samples based on Bray–Curtis distance matrices by QIIME. Adonis analysis was used to further assess the diversity differences between the two habitats by R statistical software (R packages "vegan").

We used linear discriminant analysis effect size (LEfSe) [52] to identify the taxonomic groups with significant differences in abundance in different tissues. This test uses the nonparametric factorial Kruskal–Wallis rank–sum test to detect discriminant features (taxonomic groups) with significantly differential abundance between tissues. The biological significance of these features was subsequently investigated by completing pairwise tests between the abundance of the selected features using an unpaired Wilcoxon rank–sum test. Finally, for the LEfSe, we used the linear discriminant analysis (LDA) score to quantify the effect size of each differentially abundant feature. The analyses were performed with the default values on all parameters. The threshold for the LDA parameter was 3. All data were analyzed using the Majorbio Cloud Platform (https://cloud.majorbio.com/, accessed on 17 October 2022).

We used Spearman's rank test for correlation tests between the independent variables (water temperature, air temperature, humidity, water pH, water depth, different habitats, and anthropogenic disturbance) and dependent variables (alpha diversity of skin bacterial communities, and relative abundance of dominant phyla and families). Stepwise regression was used to examine the effects of multiple independent variables on the dependent variables. The two tests were completed using SPSS software.

## 3. Results

### 3.1. Composition of Skin Microbiota in Maoershan Hynobiids

The ASVs (n = 11,352) obtained from the samples consisted of 58 phyla, 682 families, and 1568 microbial genera. The top four phyla were Proteobacteria (70.85 ± 23.46%), Firmicutes (10.12 ± 15.54%), Bacteroidota (8.00 ± 15.91%), and Actinobacteriota (5.42 ± 7.45%). At the family level, 682 bacterial groups were obtained, and the dominant family was Comamonadaceae (23.32 ± 24.22%), followed by Pseudomonadaceae (17.32 ± 16.49%), Burkholderiaceae (11.48 ± 16.97%), and Nocardiaceae (4.72 ± 7.55%; Figure 2).

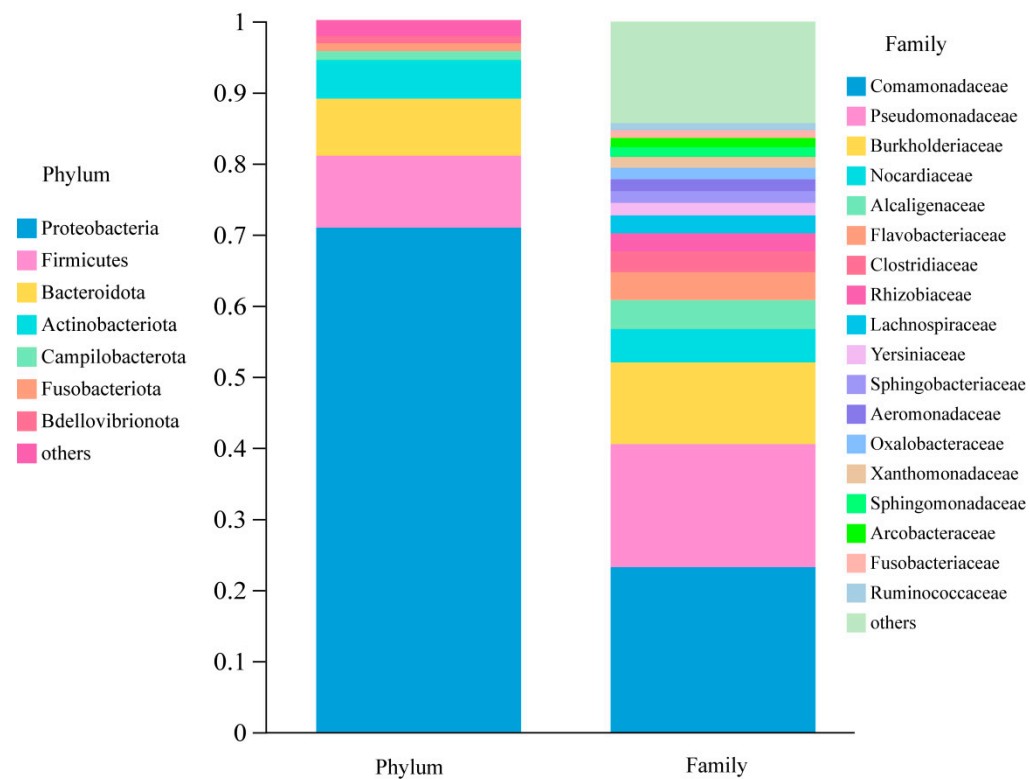

**Figure 2.** Composition of the skin microbiota at the phylum and family levels. All taxa with a relative abundance of less than 1% were classified as "others".

### 3.2. Alpha and Beta Diversity of the Skin Microbiota Varies in Habitats

The ACE index (305.347 ± 448.983), Chao index (298.747 ± 438.183), Shannon index (2.448 ± 1.156), and Simpson index (0.243 ± 0.158) in the skin microbiota of all samples were obtained. Furthermore, the richness and diversity of the skin microbiota in adults did not differ significantly between the HADD habitats and LADD habitats (Table 1, Figure 3A). In addition, the skin microbiota diversity in the hindlimb bud-stage tadpoles did not differ significantly between the HADD habitats and LADD habitats (Table 1, Figure 3B). The richness of the skin microbiota in the forelimb bud-stage tadpoles did not differ between the HADD habitats and LADD habitats (Table 1, Figure 3C). However, there were significant differences in the ACE index of the hindlimb bud-stage tadpoles between the HADD habitats and LADD habitats (Table 1, Figure 3B). Similarly, the Chao index diverged between the HADD habitats and LADD habitats in the hindlimb bud-stage tadpoles (Table 1, Figure 3B). Moreover, there were significant differences in the Shannon index in the forelimb bud-stage tadpoles between the HADD habitats and LADD habitats (Table 1, Figure 3C). The Simpson index of the forelimb bud-stage tadpoles also varied highly significantly between the HADD habitats and LADD habitats (Table 1, Figure 3C). According to the Bray−Curtis distance, the beta diversity of the skin microbiota of adults ($R^2$ = 0.125, $p$ = 0.005), hindlimb bud-stage tadpoles ($R^2$ = 0.168, $p$ = 0.037), and forelimb bud-stage tadpoles ($R^2$ = 0.220, $p$ = 0.001) exhibited significant habitat separation (Figure 4).

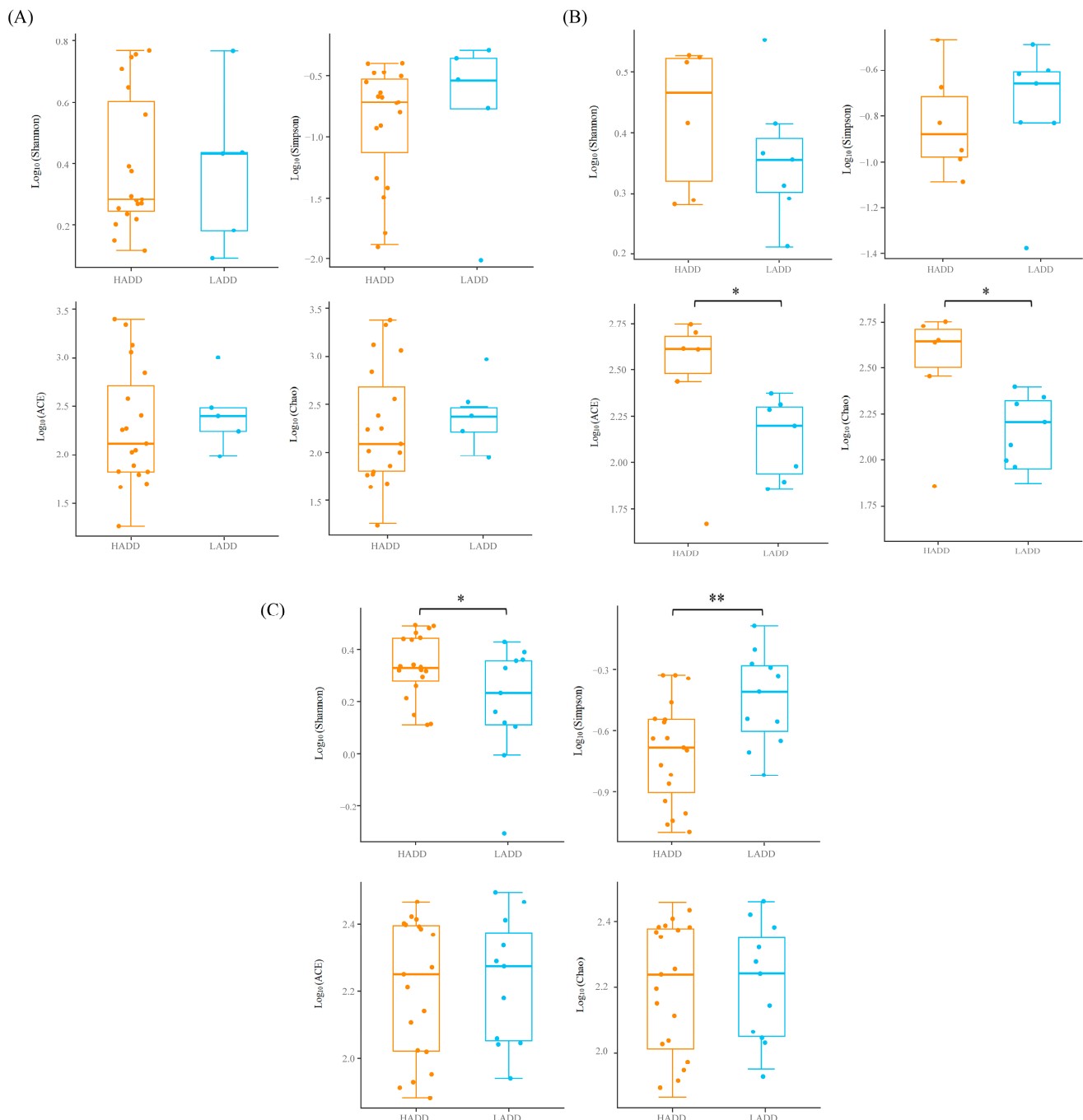

**Figure 3.** Alpha diversity indices of the skin microbiota of adult Maoershan hynobiids (**A**), hindlimb bud-stage tadpoles (**B**), forelimb bud-stage tadpoles (**C**). Orange indicates the HADD habitat, and light blue indicates the LADD habitat; "*" represents a significant difference, and "**" indicates a highly significant difference; a dot indicates a single sample.

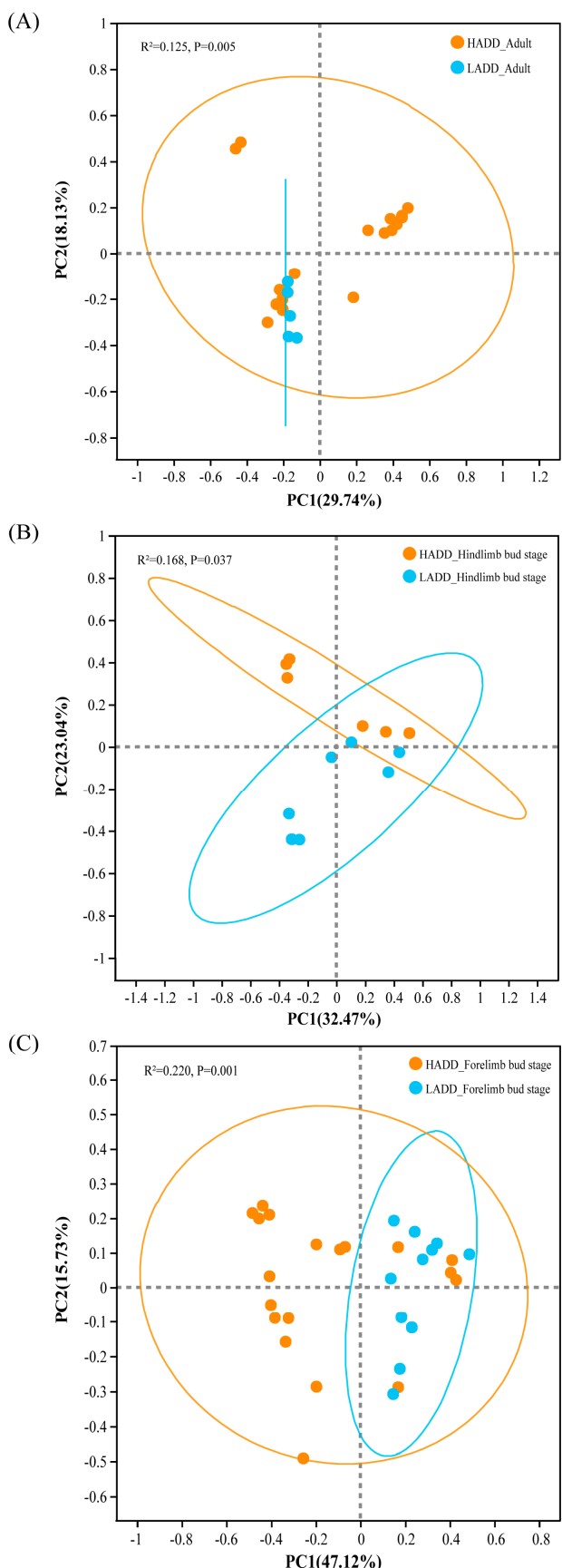

**Figure 4.** Comparison of the beta diversity of the skin microbiota among adults (**A**), hindlimb bud-stage tadpoles (**B**), and forelimb bud-stage tadpoles (**C**) based on ASVs (tested by Adonis), a dot indicates a single sample.

**Table 1.** Alpha diversity indices of the skin microbiota of Maoershan hynobiids.

| Estimators | HADD | LADD | Z/t | n/df | *p*-Value |
|---|---|---|---|---|---|
| | Mean ± SD | Mean ± SD | | | |
| Adult | | | | | |
| ACE | 522.735 ± 760.209 | 375.647 ± 339.347 | Z = −0.675 | n = 24 | 0.499 |
| Chao | 512.983 ± 740.024 | 373.359 ± 339.983 | Z = −0.675 | n = 24 | 0.499 |
| Shannon | 2.854 ± 1.584 | 2.811 ± 1.635 | Z = −0.107 | n = 24 | 0.915 |
| Simpson | 0.193 ± 0.125 | 0.282 ± 0.179 | Z = −0889 | n = 24 | 0.374 |
| Hindlimb bud tadpole | | | | | |
| ACE | 366.303 ± 186.993 | 148.357 ± 66.579 | t = 2.895 | df = 11 | 0.015 |
| Chao | 355.946 ± 177.974 | 146.026 ± 65.794 | t = 2.915 | df = 11 | 0.014 |
| Shannon | 2.864 ± 0.709 | 2.415 ± 0.648 | t = 1.097 | df = 11 | 0.296 |
| Simpson | 0.167 ± 0.0881 | 0.197 ± 0.085 | t = −0.578 | df = 11 | 0.575 |
| Forelimb bud tadpole | | | | | |
| ACE | 177.763 ± 73.048 | 184.933 ± 74.466 | t = −0.249 | df = 28 | 0.806 |
| Chao | 172.613 ± 70.407 | 178.643 ± 73.244 | t = −0.215 | df = 28 | 0.831 |
| Shannon | 2.244 ± 0.601 | 1.731 ± 0.688 | t = 2.138 | df = 28 | 0.041 |
| Simpson | 0.231 ± 0.129 | 0.407 ± 0.205 | t = −2.900 | df = 28 | 0.007 |

### 3.3. Differences in the Skin Microbiota between HADD Habitats and LADD Habitats

LEfse was used to detect differences in the abundance of microbes at different levels (phylum, class, order, family, and genus) to further identify shifts in composition in different habitats and hosts. In adults, the results revealed 137 skin bacterial taxa with differential abundance between both habitats, including 118 taxa from LADD habitats and 19 taxa from HADD habitats. The phylum Proteobacteria, class Gammaproteobacteria, phylum Bacteroidota, and class Bacteroidia were the major taxa contributing to these differences (Figure S1). In hindlimb bud-stage tadpoles, the results revealed 34 skin bacterial taxa with differential abundance between the habitats, of which 7 taxa were from LADD habitats and 27 were from HADD habitats. The main groups responsible for these differences were the class Bacteroidia, phylum Bacteroidota, family Aeromonadaceae, and order Aeromonadales (Figure S2). In forelimb bud-stage tadpoles, 51 skin bacterial taxa had substantial differences in abundance between the HADD habitats and LADD habitats, of which 23 taxa were from LADD habitats and 28 were from HADD habitats. The genus Limnohabitans, families Comamonadaceae and Burkholderiaceae, and genus *Polynucleobacter* were the major taxa responsible for these differences (Figure S3).

### 3.4. Effects of Ecological Factors on Skin Microbiota Diversity and Dominant Bacteria

Spearman's rank test demonstrated that ecological factors and anthropogenic disturbance were significantly correlated with the alpha diversity of the skin microbiota (Tables S2–S4). In adults, there was a significant correlation between skin microbiota richness and anthropogenic disturbance (ACE, r = −0.542, n = 24, *p* = 0.006, Chao, r = −0.541, n = 24, *p* = 0.006). In addition, the alpha diversity of the skin microbiota was significantly correlated with ecological factors (Table S2). In hindlimb bud-stage tadpoles, skin microbiota richness was positively correlated with anthropogenic disturbance (ACE, r = 0.642, n = 13, *p* = 0.018, Chao, r = 0.642, n = 13, *p* = 0.018), and the correlation between alpha diversity and ecological factors was reflected in water temperature and different habitats (Table S3). In forelimb bud-stage tadpoles, skin microbiota diversity was significantly correlated with anthropogenic disturbance (Shannon, r = 0.460, n = 30, *p* = 0.010, Simpson, r = −0.592, n = 30, *p* = 0.001), and the alpha diversity of the microbiota was significantly associated with six ecological factors (Table S4). Additionally, the relative abundance of dominant bacteria was also significantly correlated with anthropogenic disturbance and ecological factors (such as temperature, pH, and humidity; Tables S2–S4).

The results of stepwise regression analysis illustrate that of the seven factors, water depth was the main factor affecting the richness of the skin microbiota in adults, whereas

air temperature and water temperature were the main factors affecting the skin microbiota diversity of adults (Table 2). The skin microbiota richness decreased as the water depth increased (ACE: r = −0.732, df = 23, $p < 0.001$, Chao: r = −0.731, df = 23, $p = 0.004$). With increasing air temperature, the Shannon index decreased (Shannon: r = −0.690, df = 23, $p < 0.001$), whereas the Simpson index increased with increasing water temperature (Simpson: r = 0.578, df = 23, $p = 0.003$). In adults, the four most dominant phyla were Proteobacteria, Bacteroidota, Actinobacteriota, and Firmicutes, whereas the four dominant families were Pseudomonadaceae, Flavobacteriaceae, Alcaligenaceae, and Nocardiaceae. The relative abundance of the dominant bacterial phyla and families was affected by anthropogenic disturbance and ecological factors (including water pH, air temperature, and different habitats). For example, the relative abundance of Actinobacteriota, Pseudomonadaceae, Alcaligenaceae, and Nocardiaceae decreased with increasing water pH, the richness of Proteobacteria increased with increasing anthropogenic disturbance, and the relative abundance of Bacteroidota increased with decreasing anthropogenic disturbance. Stepwise regression analysis indicated that the relative abundance of Firmicutes was not affected by anthropogenic disturbance or ecological factors (Table 2).

**Table 2.** Effects of specific factors on the alpha diversity of the skin microbiota and relative abundance of dominant bacteria in adult Maoershan hynobiids based on stepwise regression analysis.

| Dependent Variable | Detectable Regression Factor | Regression Coefficient | | |
|---|---|---|---|---|
| | | r | df | p |
| ACE | Water depth | −0.732 | 23 | <0.001 |
| Chao | Water depth | −0.731 | 23 | 0.004 |
| Shannon | Air temperature | −0.690 | 23 | <0.001 |
| Simpson | Water temperature | 0.578 | 23 | 0.003 |
| Proteobacteria | Anthropogenic disturbance | 0.764 | 23 | <0.001 |
| Bacteroidota | Anthropogenic disturbance | −0.522 | 23 | 0.009 |
| Actinobacteriota | Water pH | −0.490 | 23 | 0.015 |
| Firmicutes | | | | |
| Pseudomonadaceae | Water pH | −1.618 | 23 | <0.001 |
| Pseudomonadaceae | Air temperature | −0.974 | 23 | 0.017 |
| Alcaligenaceae | Water pH | −0.561 | 23 | 0.004 |
| Nocardiaceae | Water pH | −0.536 | 23 | 0.007 |
| Flavobacteriaceae | Different habitats | −0.518 | 23 | 0.01 |

In the hindlimb bud stage, the richness of the skin microbiota was significantly affected by anthropogenic disturbance, and the skin microbiota diversity was affected by the water pH (Table 3). When anthropogenic disturbance increased, the skin microbiota richness increased (ACE: r = 0.722, df = 12, $p = 0.005$, Chao: r = 0.737, df = 12, $p = 0.004$), whereas the skin microbiota diversity decreased with increasing water pH (Shannon: r = −0.717, df = 12, $p = 0.006$, Simpson: r = 0.661, df = 12, $p = 0.014$). In hindlimb bud-stage tadpoles, Proteobacteria, Firmicutes, Actinobacteriota, Fusobacterio, and Comamonadaceae were the main phyla affected by ecological factors, whereas Pseudomonadaceae, Pseudomonadaceae, and Clostridiaceae were the main families affected by ecological factors. The relative abundance of the dominant bacterial phyla was influenced by the water temperature, water depth, anthropogenic disturbance, and habitat. The relative abundance of the dominant families was influenced by anthropogenic disturbance and water depth. Stepwise regression analysis indicated that the relative abundance of Proteobacteria, Burkholderiaceae, Pseudomonadaceae, and Aeromonadaceae was not affected by anthropogenic disturbance or ecological factors (Table 3).

**Table 3.** Effects of specific factors on the alpha diversity of the skin microbiota and relative abundance of dominant bacteria in hindlimb bud-stage Maoershan hynobiid tadpoles based on stepwise regression analysis.

| Dependent Variable | Detectable Regression Factor | Regression Coefficient | | |
| --- | --- | --- | --- | --- |
| | | r | df | *p* |
| ACE | Anthropogenic disturbance | 0.722 | 12 | 0.005 |
| Chao | Anthropogenic disturbance | 0.737 | 12 | 0.004 |
| Shannon | Water pH | −0.717 | 12 | 0.006 |
| Simpson | Water pH | 0.661 | 12 | 0.014 |
| Proteobacteria | | | | |
| Bacteroidota | Different habitats | −0.647 | 12 | 0.017 |
| Actinobacteriota | Water depth | −1.020 | 12 | <0.001 |
| Actinobacteriota | Anthropogenic disturbance | 0.447 | 12 | 0.018 |
| Campylobacterota | Water pH | −0.719 | 12 | <0.001 |
| Campylobacterota | Anthropogenic disturbance | 0.384 | 12 | 0.002 |
| Campylobacterota | Water temperature | −0.210 | 12 | 0.025 |
| Comamonadaceae | Water depth | −1.025 | 12 | <0.001 |
| Comamonadaceae | Anthropogenic disturbance | 0.372 | 12 | 0.033 |
| Burkholderiaceae | | | | |
| Pseudomonadaceae | | | | |
| Aeromonadaceae | | | | |

In the forelimb bud stage, the skin microbiota richness was significantly influenced by humidity, whereas anthropogenic disturbance affected the skin microbiota diversity (Table 4). Skin microbiota richness increased with increasing humidity (ACE: r = 0.561, df = 29, *p* = 0.001, Chao: r = 0.562, df = 29, *p* = 0.001); furthermore, microbiota diversity increased with increasing anthropogenic disturbance (Shannon: r = 0.566, df = 29, *p* < 0.001, Simpson: r = −0.607, df = 29, *p* < 0.001). Similar to other stages, the main families affected by ecological factors in forelimb bud-stage tadpoles were Proteobacteria, Firmicutes, Actinobacteriota, and Fusobacterio, whereas the main affected families were Comamonadaceae, Pseudomonadaceae, Burkholderiaceae, and Clostridiaceae. The relative abundance of the families such as Burkholderiaceae and Pseudomonadaceae increased with increasing anthropogenic disturbance, whereas the relative abundance of Comamonadaceae increased with decreasing anthropogenic disturbance. The relative abundance of dominant phyla such as Actinobacteriota increased with decreasing humidity. Stepwise regression analysis indicated that the relative abundance of Firmicutes, Fusobacteriota, Proteobacteria, and Clostridiaceae was not affected by anthropogenic disturbance or ecological factors (Table 4).

**Table 4.** Effects of specific factors on the alpha diversity of the skin microbiota and relative abundance of dominant bacteria in forelimb bud-stage Maoershan hynobiid tadpoles based on stepwise regression analysis.

| Dependent Variable | Detectable Regression Factor | Regression Coefficient | | |
| --- | --- | --- | --- | --- |
| | | r | df | *p* |
| ACE | Humidity | 0.561 | 29 | 0.001 |
| Chao | Humidity | 0.562 | 29 | 0.001 |
| Shannon | Anthropogenic disturbance | 0.566 | 29 | 0.001 |
| Simpson | Anthropogenic disturbance | −0.607 | 29 | <0.001 |
| Actinobacteriota | Humidity | −0.932 | 29 | <0.001 |
| Firmicutes | | | | |
| Fusobacteriota | | | | |
| Proteobacteria | | | | |
| Pseudomonadaceae | Anthropogenic disturbance | 0.613 | 29 | <0.001 |
| Comamonadaceae | Anthropogenic disturbance | −0.756 | 29 | <0.001 |
| Burkholderiaceae | Anthropogenic disturbance | 0.855 | 29 | <0.001 |
| Clostridiaceae | | | | |

## 4. Discussion

### 4.1. Characteristics of the Skin Microbiota of Maoershan hynobiids

Proteobacteria and Firmicutes dominated the skin microbiota in Maoershan hynobiids, followed by Bacteroidota and Actinobacteriota. Similar to the reports for other amphibians, the dominant microbial phyla of *Isthmohyla pseudopuma* and *Agalychnis callidryas* were Proteobacteria, Firmicutes, Bacteroidota, and Actinobacteriota (Table 5). Different species have differences in the dominant microbial phyla. For example, the dominant phyla in the skin microbiotas of frogs (*Rana sylvatica*) inhabiting the forest ground are Proteobacteria, Bacteroidetes, Actinobacteria, and Verrucomicrobiota (Table 5). Verrucomicrobia species are commonly found in the gut [53] and soil [54]. The dominant bacteria in the Maoershan hynobiid skin microbiota did not include Verrucomicrobiota. The differences in the dominant phyla of the amphibian skin microbiota might be related to their habitats, and microbial communities might be selected from the environment by the amphibian host [18].

**Table 5.** The top four phyla of amphibian skin microbiota.

| Species | Locations | Dominant Bacteria Phyla | Habitat | References |
|---|---|---|---|---|
| *Craugastor fitzingeri* (Anura) | Heredia province, Costa Rica | Proteobacteria, Bacteroidetes, Actinobacteria, Acidobacteria | Terrestrial | [55] |
| *Isthmohyla pseudopuma* (Anura) | Alajuela province, Costa Rica | Proteobacteria, Firmicutes, Bacteroidetes, Actinobacteria | Ponds | [15] |
| *Agalychnis callidryas* (Anura) | Panamá province, Panamá | Proteobacteria, Actinobacteria, Firmicutes, Bacteroidetes | Ponds | [8] |
| *Rana sylvatica* (Anura) | Ontario, Canada | Proteobacteria, Bacteroidetes, Actinobacteria, Verrucomicrobiota | Ponds and forest floor | [16] |
| *Andrias davidianus* (Urodela) | Sichuan province, China | Proteobacteria, Firmicutes, Bacteroidetes, Verrucomicrobiota | Artificial habitats | [56] |
| *Hynobius maoershanensis* (Urodela) | Guangxi, China | Proteobacteria, Firmicutes, Bacteroidota, Actinobacteriota | Deep pool and lake | This research |

Concerning the predominant bacterial phyla, Proteobacteria are typical dominant phyla in amphibian skin microbiota [57,58] and related to amphibian defense and immunity [16,35]. Members of Proteobacteria have extremely diverse morphologies and physiological functions, and a competitive advantage for survival in various ecological environments [59]. Some bacteria in Proteobacteria can help the host inhibit pathogens. For example, Sphingomonadales, Pseudomonadales, and Burkholderiales inhibit Bd, and they are negatively correlated with the growth of dermatophytes [60]. In our study, Sphingomonadales and Burkholderiales were significantly enriched in the skin microbiota in Maoershan hynobiid tadpoles. These families might function by helping tadpoles defend against fungal threats. Firmicutes also helps to inhibit pathogens. For example, the proportion of Firmicutes in the skin of healthy giant salamanders decreased sharply after being disturbed by pathogenic bacteria, indicating an antagonistic relationship between Firmicutes and pathogenic bacteria [61]. The relative abundance of Proteobacteria and Firmicutes reflects the ability of the host to resist pathogens.

### 4.2. Effects of Ecological Factors on the Skin Microbiota

The environment regulates the growth of microorganisms by affecting their affinity for the substrate [62] or the chemical content of amphibian skin [63]. Microorganisms are similar to other organisms, as they have optimal temperatures for growth and reproduction [64]. Studies have revealed that temperature is significantly correlated with the skin microbiota diversity and richness of [64]. Consistent with our second prediction, ecological factors influence skin microbiota diversity and the relative abundance of dominant bacteria. In our study, air temperature and water temperature affected the skin microbiota diversity and the relative abundance of dominant bacteria. With increasing air temperature and water temperature, the diversity of the microbiota and the relative abundance of the dominant

bacteria decreased. Temperature can alter the microbial communities in the skin by preventing the growth of certain bacterial species [65]. It has been reported that temperature affects the molting frequency of the cane toad (*Rhinella marina*), which then affects the colonization and growth of the skin microbiota [66].

　　Humidity probably affects the production of antimicrobial peptide secretions by the host skin, which subsequently affects the skin microbiota diversity [24,25]. Humidity had a significant effect on the skin microbiota richness and the relative abundance of dominant bacteria in the forelimb bud stage of Maoershan hynobiids. Skin microbiota richness increased with increasing humidity, but the relative abundance of the dominant phylum Actinobacteriota decreased. Furthermore, increasing humidity in the environment could favor a wider range of activities for amphibians, thus affecting the diversity of the skin microbiota [67,68]. For example, frogs can expand their range of foraging activities during the rainy season; consequently, some bacteria became relatively more abundant, whereas some bacterial categories decreased [68]. We observed the activity of the Maoershan hynobiids in the rainy season when the humidity was extremely high. The contact of the skin of the Maoershan hynobiids with the road had a certain impact on the skin microbiota (Figure 5).

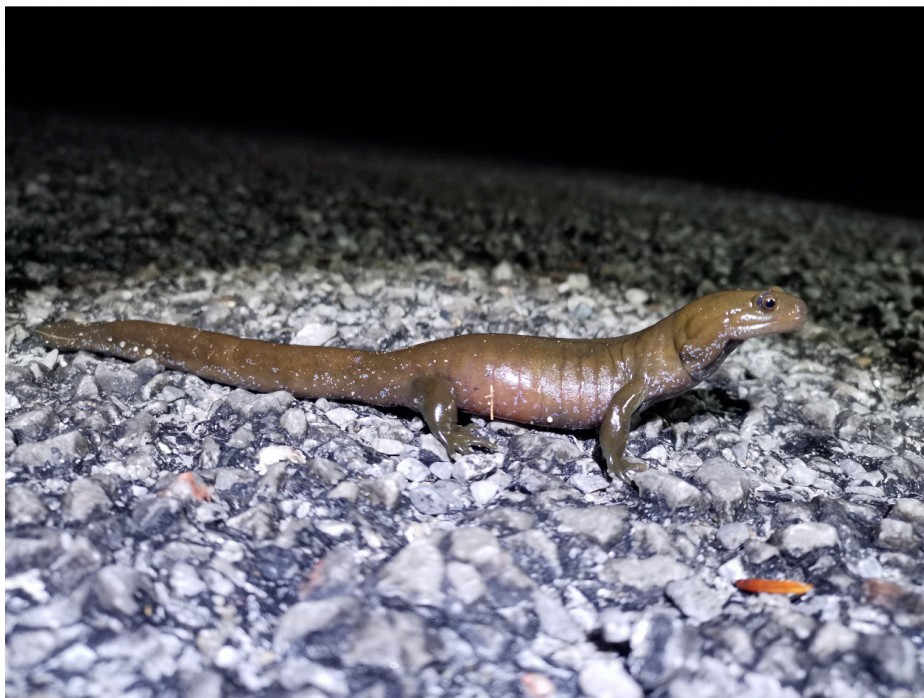

**Figure 5.** Adult Maoershan hynobiid moving on the road.

　　Water pH affects the diversity of the skin microbiota, possibly by affecting the secretion of antimicrobial peptides by amphibian skin [24,25]. Our results demonstrate that the skin microbiota diversity and relative abundance of the dominant bacteria were affected by the water pH. With increasing pH, the skin microbiota diversity and relative abundance of dominant bacteria decreased. Previous studies revealed that the diversity of the axolotl skin microbiota is affected by changes in water pH [69]. In addition, the microbial community structure of frog skin is influenced by soil pH, as the alpha diversity is highest in sites with lower pH [68]. Furthermore, the soil pH can shape the spatial layout of soil bacterial communities [70]. This might explain the differences in microbial community structure when a salamander (*Plethodon cinereus*) can acquire its bacteria in the environment [20].

　　Previous studies found that the size of water bodies affects the structure of the host skin microbiota [63]. Similarly, the depth of water influenced the skin microbiota richness in adults in our study. With increasing water depth, the richness of the skin microbiota of

the Maoershan hynobiids decreased. However, the specific effect of water depth on the skin microbiota of Maoershan hynobiids needs to be further studied.

*4.3. Effects of Anthropogenic Disturbance on the Skin Microbiota*

Anthropogenic disturbance might affect the diversity of the amphibian skin microbiota [37]. Our results indicate that anthropogenic disturbance significantly affects the skin microbiota of Maoershan hynobiid. Alpha diversity during the tadpole period was significantly higher in HADD habitats than in LADD habitats, contrary to the first prediction. Anthropogenic disturbance was the main factor affecting the skin microbiota alpha diversity, which supports the second prediction. The egg bag generally hatches into the hindlimb bud stage in July or August [71], which is the peak tourist season and the period of greatest anthropogenic disturbance. Tourist activities might lead to abnormal corticosterone levels in the hosts in HADD habitats, and the accumulated content could fuel the growth of pathogenic microorganisms [34,72]. For example, corticosterone levels in the northern leopard frog increased with increasing disturbance by human activity [72]. At the same time, the individual body structure of Maoershan hynobiids greatly changes in this stage, and skin shedding and remodeling occur during abnormal development, possibly forming an open niche [60] and making the animal more susceptible to interference. The effect of habitat disturbance on the skin microbiota of each species is different [73], and our study shows that habitat disturbance increased the diversity of the skin microbiota. There have been some studies were habitat disturbance reduced the diversity of skin microbiota, for example, habitat disturbance reduced skin microbiota diversity in neotropical-montane frog tadpoles, which could increase their susceptibility to pathogens and lead to adverse effects on their health status [34]. The reason for this difference may be the species differences or the different interference indicators adopted [43,73]. In this study, we could not determine whether the increased diversity of the skin microbiota in the disturbed habitat was harmful or beneficial.

In line with our second prediction, the results of stepwise regression analysis indicate that anthropogenic disturbance was the main factor affecting the relative abundance of the dominant microbiota. Anthropogenic disturbance alters the abundance of probiotics and pathogenic bacteria in the environment [74], which is potentially pathogenic to amphibians [75]. The results of this study indicate that anthropogenic disturbance affected the dominant microbes. For example, anthropogenic disturbance had a negative effect on the abundance of Actinobacteria in the skin microbiota in the adult stage. Actinobacteria can produce antibiotics, which can effectively inhibit most pathogenic microorganisms [76]. The negative impact of anthropogenic disturbance on the relative abundance of Actinobacteria might be detrimental to the antimicrobial ability of Maoershan hynobiids' skin microbiota. Moreover, the relative abundance of pathogenic bacteria, such as Flavobacteriaceae, was influenced by the habitat type. Aeromonadaceae had a significant positive correlation with anthropogenic disturbance, and it was significantly enriched in HADD habitats. Aeromonadaceae has been reported as an important group of pathogens in amphibians [77]. The enrichment of Aeromonadaceae in HADD habitats could pose a potential threat to the health of Maoershan hynobiids. Previous studies revealed that anti-pathogenic bacteria in the frog skin microbiota are affected by human activity-induced forest fragmentation, and these changes might have adverse effects on the health of the host [1]. Disturbance caused by human activity can shape the skin microbiota of amphibians [43]. For example, chemical contamination caused by humans reduces the skin microbiota diversity of amphibians, specifically affecting the structure and composition of their microbiota [33]. In addition, the distance of roads from habitats can greatly affect the skin microbiota of amphibians [78]. Our results show that the skin microbiota richness of Maoershan hynobiids in HADD habitats was greater than LADD habitats. This may be because the HADD habitats were closer to the road than LADD habitats. Similarly, in one study, eastern newts closer to the road had greater skin microbiota richness than those far from the road [78]. In aquatic and terrestrial habitats, road-related salinization and eutrophication may influence the

community structure of microbes in the environment [79]. For example, road salts may lead to increased halotolerant bacteria in the environment, which may affect the aquatic communities [79,80]. In this study, road salts were also distributed in winter, but whether this affects the aquatic microbial communities needs further research.

## 5. Conclusions

Our results show significant differences in the alpha diversity of the skin microbiota of Maoershan hynobiids between HADD habitats and LADD habitats. The greater diversity and richness of the skin microbiota in the HADD habitats could facilitate the adaptation of Maoershan hynobiids to disturbed environments. The diversity and richness of the skin microbiota were affected by anthropogenic disturbance, as both increased with increasing anthropogenic disturbance. The diversity of Maoershan hynobiids' skin bacterial communities and the dominant bacteria were affected by ecological factors. Skin microbiota diversity and the relative dominant bacterial abundance decreased with increasing temperature and pH. Our results broaden our understanding of the skin microbiota of amphibians. These findings will also be helpful for understanding the relationship between the skin microbial community composition and environmental factors. It has not been verified that the diversity and abundance of skin microbiota in HADD habitats have a negative impact on species survival; however, as time increases, the accumulation of interference may affect the survival of Maoershan hynobiids. Based on this study, we believe that it is necessary to control the distance between the activity range of tourists and the habitat of Maoershan hynobiids at a suitable distance.

**Supplementary Materials:** The following supporting information can be downloaded at: https://www.mdpi.com/article/10.3390/d15080932/s1, Figure S1: The LEfSe of the skin microbiota abundance of adult Maoershan hynobiids in HADD and LADD habitats; Figure S2: The LEfSe of the skin microbiota abundance of hindlimb bud-stage tadpoles in HADD and LADD habitats; Figure S3: The LEfSe of the skin microbiota abundance of forelimb bud-stage tadpoles in HADD and LADD habitats; Table S1: Data of ecological factors and anthropogenic disturbance in Maoershan hynobiid habitats (the values of each factor are the mean). Table S2: Results of the Spearman's rank test between skin microbiota of adult and ecological factors and anthropogenic disturbance; Table S3: Results of the Spearman's rank test between skin microbiota of hindlimb bud-stage tadpoles and ecological factors and anthropogenic disturbance; Table S4: Results of the Spearman's rank test between skin microbiota of forelimb bud-stage tadpoles and ecological factors and anthropogenic disturbance

**Author Contributions:** H.C.: Data curation (lead); investigation (equal); validation (equal); writing—original draft (lead); Y.H.: investigation (equal); G.P.: investigation (equal); Z.C.: investigation (equal); Z.W.: funding acquisition (equal); methodology (equal); resources (equal). H.H.: conceptualization (equal); funding acquisition (equal); project administration (equal); supervision (equal); visualization (equal); writing—review and editing (equal). All authors have read and agreed to the published version of the manuscript.

**Funding:** This research was funded by the National Natural Science Foundation of China, grant number 31860609, the Guangxi Natural Science Foundation project, grant number 2023GXNSFAA026501, and the Innovation Project of Guangxi Graduate Education, China, grant number YCSW2023152.

**Institutional Review Board Statement:** All animal procedures were authorized by the Institutional Animal Care and Use Committee and approved by the Laboratory Animal Care and Animal Ethics Committee of Guangxi Normal University (No. 202301-001).

**Informed Consent Statement:** Not applicable.

**Data Availability Statement:** The datasets presented in this study can be found in online repositories. The names of the repository/repositories and accession number(s) can be found here: https://www.ncbi.nlm.nih.gov/, PRJNA935520.

**Acknowledgments:** We are very grateful to the Guangxi Maoershan National Nature Reserve for allowing us to conduct research at the site. We appreciate the assistance of Ye Jianping of the Maoershan National Nature Reserve in Guangxi. We also thank the members of the Bajiaotian Management Station of the Maoershan National Nature Reserve in Guangxi for their assistance in the field.

**Conflicts of Interest:** The authors declare that the research was conducted with the absence of any commercial or financial relationships that could be construed as a potential conflict of interest.

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
