# Peer review of "Ecological Factors and Anthropogenic Disturbance May Restructure the Skin Microbiota of Maoershan Hynobiids (Hynobius maoershanensis)"

_diversity, doi:10.3390/d15080932_

Round 1
Reviewer 1 Report
The work is well written and the subject is interesting and very topical. Just a few comments about the number of sites for comparison. How do authors justify the use of only two sites of contrasting conditions. Although the demonstrate the anthropogenic effect on bacterial communities, they do not present species implications for conservation or suggestions for the protection of H. maoershanensis
Author Response
Thank you for your comments. The relevant information about this species is scarce, and it is a strictly narrow domain species, which has been found in several places. The starting point of this article is to see the influence of human disturbance on it. A site has obvious human disturbance, and a site has minimal human disturbance. These two places are currently relatively typical, so only these two sites were selected for the study. We suggest the conservation at the conclusions of the article.
Reviewer 2 Report
The “Ecological Factors and Anthropogenic Disturbance May Restructure the Skin Microbiota of the Maoershan Hynobiids (Hynobius maoershanensis)” presents a very well designed study exploring the effect of environment on skin microbial communities in a Caudata. The authors present a generally well written manuscript rich on methodological details. While I enjoyed the manuscript very much, I have two issues described below. Additionally, I provide line-by-line comments with regards to minor things.
Issue1: Authors introduce a lot of environmental factors that vary between habitats, however at any point they state which factor is expected to be higher or lower (one exception can be found in the anthropogenic disturbance). I believe the authors need to state in the methods (perhaps as a supplementary table) how these parameters vary between sites (which site has higher pH, temp, depth, …).
Issue2: From the manuscript it seems like the authors could have sampled the same individuals over and over again during the 1.5 year sampling period. If the same individual was sampled more than once, that information should be included into the analysis. Do authors have a way check this? Independently of the answer to my questions, this issue should be addressed in the manuscript as it could lead to different conclusions.
Line by line:
Line 19-22: Authors describe the findings in two of the three lifestage. Consider adding the information for adults as well to reflect more clearly the study.
Line 24: Clarify that it is air temperature.
Line 24-26: The first clause of this sentence is a repetition of what was said above (“The diversity of the skin microbiota in forelimb bud stage tadpoles from HADD habitats was higher than that in their counterparts from LADD habitats. The richness of the skin microbiota in hindlimb bud stage tadpoles was greater in HADD habitats than in LADD habitats.” == “In addition, the diversity and richness of the skin microbiota increased with increasing anthropogenic disturbance”). Please remove one of them.
Line 30-31: This is an inappropriate choice of key words as by being the same words as the title they contribute nothing to increased discoverability of the paper. I venture some suggestions: Microbiome, Cutaneous bacterial communities, Amphibia, Caudata, 16S amplicon sequencing.
Line 44: Comma missing before “or possibly from…”
Line 45-47: Unclear what the authors are trying to convey. I believe authors wanted to say that the mucus glands are what keeps the amphibians moist. If that is the case replace the “and” after the comma by “as”.
Line 48: I would suggest putting the examples between brackets to increase readability.
Line 66: Please add the oxford comma in the brackets
Line 99: Please add the thousand separator to all numbers over 999 throughout the manuscript.
Line 112-117: Very long sentence. Consider splitting it in two.
Line 131-132: Unclear what the authors are trying to say. Potentially “where” was supposed to be “was”.
Line 135: Oxford comma missing before “and”
Line 137: Oxford comma missing before “and”
Line 147: “Frequent”, not “frequently”
Line 201: Please cite Mothur
Line 202: Oxford comma missing before “and”
Line 203-206: Which software/packages were used to performed these analysis (Kolmogorov Smirnov test, t-test, Mann-Whitney U Test)? Please clarify that in the text and cite the software/packages where needed.
Line 209: Please cite R. Where there any packages used to create these graphs? If so please cite them.
Line 210-211: No need to cite the QIIME2 version here as it was done above.
Line 211: How (Software/packages) was this analysis (Adonis analysis) performed?
Line 220-221: Authors only specify the LDA value used, what about all other parameters? Assuming only default parameters were used I would recommend altering this sentence to “Analyses were performed with default values on all parameters.”
Line 221: Please cite Majorbio Cloud Platform
Line 234: Text seems to contradict itself. In the previous sentence the authors say that there are 682 families, but now it says 386. Please clarify the reason behind this apparent disconnect.
Line 241-263: I would strongly recommend moving all statistical values to a single table. As it is this paragraph is very difficult to read and statistical data is split across the paragraph.
Line 275-277: Authors appear to have made a random selection of which groups to list. For instance, why is Bacteroidota listed, but not Bacteroidia (which has the same LDA score)? I would suggest either using an LDA threshold (preferred) or just listing the top two of each habitat. This was interestingly done for the other two life-stages, so only adults need to be adjusted.
Line 287-301: Table numbers do not match table numbers in the supplementary file. Additionally, this is not the notation of the journal for supplementary materials, it should be simply S1, S2, and S3.
Line 290: Oxford comma missing before “and”
Line 367: Species name not italicized
Line 379: Oxford comma missing before “and”.
Line 393-405: I am not pleased with this explanation (specially with using an example of richness to support a difference found in diversity). Authors found a pattern that contradicts what is known from literature. While species differences might be the reason for it, it is too simple of an explanation. I would like to know whether the level of disturbance between these studies is equivalent. For instance, are the disturbed areas in the previous studies agriculture related (with a lot of pesticides that kill a lot of micro-organisms), or factory pollution (also kills a lot of micro-organisms)? Also, humans (tourists) carry a lot of different bacteria with them, so in this scenario it would not surprise me that a different source of disturbance would lead to contradictory results in terms of diversity.
Line 391-414: This paragraph has some repetitions (for instance Bd) and back and forwards. I would recommend streamlining the information some more.
Line 423-424: Please clarify that it was water temperature
Line 427: Species name not italicized
References:
Line 563-565: Do not capitalize every word of the title (confirm this in all references as it happens quite frequently), specially not the specific epithet.
Line 721-722: Genus name not italicized
Line 727-729: Please remove the hyperlink
Figure 2: I would suggest adding a label to the legends in the figure saying “Phyla” and “Families” for clarity.
Supplementary Tables: Species name is not italicized in the title.
While the manuscript is well written in general, some sections (see line-by-line comments above) are repetitive and/or unclear. The manuscript needs some minor editing when it comes to grammar.
Author Response
Issue1: Authors introduce a lot of environmental factors that vary between habitats, however at any point they state which factor is expected to be higher or lower (one exception can be found in the anthropogenic disturbance). I believe the authors need to state in the methods (perhaps as a supplementary table) how these parameters vary between sites (which site has higher pH, temp, depth, …).
Response 1: Thank you for your comments. We have added the information.
Issue2: From the manuscript it seems like the authors could have sampled the same individuals over and over again during the 1.5 year sampling period. If the same individual was sampled more than once, that information should be included into the analysis. Do authors have a way check this? Independently of the answer to my questions, this issue should be addressed in the manuscript as it could lead to different conclusions.
Response 2: Thank you for your comments. We sampled with the random sampling method.
Line by line:
Line 19-22: Authors describe the findings in two of the three lifestage. Consider adding the information for adults as well to reflect more clearly the study.
Response 3: Thank you for your comments. We have added the information.
Line 24: Clarify that it is air temperature.
Response 4: Thank you for your suggestion. The temperature here actually means the water temperature and the air temperature. We have revised it.
Line 24-26: The first clause of this sentence is a repetition of what was said above (“The diversity of the skin microbiota in forelimb bud stage tadpoles from HADD habitats was higher than that in their counterparts from LADD habitats. The richness of the skin microbiota in hindlimb bud stage tadpoles was greater in HADD habitats than in LADD habitats.” == “In addition, the diversity and richness of the skin microbiota increased with increasing anthropogenic disturbance”). Please remove one of them.
Response 5: Thank you for your suggestion. We have revised it.
Line 30-31: This is an inappropriate choice of key words as by being the same words as the title they contribute nothing to increased discoverability of the paper. I venture some suggestions: Microbiome, Cutaneous bacterial communities, Amphibia, Caudata, 16S amplicon sequencing.
Response 6: Thank you for your suggestion. We have revised some key words.
Line 44: Comma missing before “or possibly from…”
Response 7: Thank you for your comments. We have revised it.
Line 45-47: Unclear what the authors are trying to convey. I believe authors wanted to say that the mucus glands are what keeps the amphibians moist. If that is the case replace the “and” after the comma by “as”.
Response 8: Thank you for your comments. You are right, what We want to express is that the mucus glands are what keeps the amphibians moist. We have revised it.
Line 48: I would suggest putting the examples between brackets to increase readability.
Response 9: Thank you for your suggestion. We have revised it.
Line 66: Please add the oxford comma in the brackets
Response 10: Thank you for your comments. We have revised it.
Line 99: Please add the thousand separator to all numbers over 999 throughout the manuscript.
Response 11: Thank you for your comments. We have revised it.
Line 112-117: Very long sentence. Consider splitting it in two.
Response 12: Thank you for your comments. We have revised it.
Line 131-132: Unclear what the authors are trying to say. Potentially “where” was supposed to be “was”.
Response 13: Thank you for your comments. We have revised it.
Line 135: Oxford comma missing before “and”
Response 14: Thank you for your comments. We have revised it.
Line 137: Oxford comma missing before “and”
Response 15: Thank you for your comments. We have revised it.
Line 147: “Frequent”, not “frequently”
Response 16: Thank you for your comments. We have revised it.
Line 201: Please cite Mothur
Response 17: Thank you for your comments. We have provided the link of Mothur in parentheses.
Line 202: Oxford comma missing before “and”
Response 18: Thank you for your comments. We have revised it.
Line 203-206: Which software/packages were used to performed these analysis (Kolmogorov Smirnov test, t-test, Mann-Whitney U Test)? Please clarify that in the text and cite the software/packages where needed.
Response 19: Thank you for your comments. We have revised it. We used the SPSS software version 23.0 (SPSS Inc., Chicago, IL, USA) to performed these analysis.
Line 209: Please cite R. Where there any packages used to create these graphs? If so please cite them.
Response 20: Thank you for your comments. We have revised it. We added the packages in parentheses.
Line 210-211: No need to cite the QIIME2 version here as it was done above.
Response 21: Thank you for your comments. We have revised it.
Line 211: How (Software/packages) was this analysis (Adonis analysis) performed?
Response 22: Thank you for your comments. We have revised it.
Line 220-221: Authors only specify the LDA value used, what about all other parameters? Assuming only default parameters were used I would recommend altering this sentence to “Analyses were performed with default values on all parameters.”
Response 23: When the default value is 2, the graph presenting the results is too long, so we set the LDA value to 3.
Line 221: Please cite Majorbio Cloud Platform
Response 24: Thank you for your comments. We have revised it.
Line 234: Text seems to contradict itself. In the previous sentence the authors say that there are 682 families, but now it says 386. Please clarify the reason behind this apparent disconnect.
Response 25: We are sorry, because of our negligence of 682 written 386, after verification, the correct information is 682 families.
Line 241-263: I would strongly recommend moving all statistical values to a single table. As it is this paragraph is very difficult to read and statistical data is split across the paragraph.
Response 26: Thank you for your suggestion. We have revised it.
Line 275-277: Authors appear to have made a random selection of which groups to list. For instance, why is Bacteroidota listed, but not Bacteroidia (which has the same LDA score)? I would suggest either using an LDA threshold (preferred) or just listing the top two of each habitat. This was interestingly done for the other two life-stages, so only adults need to be adjusted.
Response 27: Thank you for your comments. We originally listed the first 4 bacteria with the largest LDA values, now we have changed to list the top two of each habitat.
Line 287-301: Table numbers do not match table numbers in the supplementary file. Additionally, this is not the notation of the journal for supplementary materials, it should be simply S1, S2, and S3.
Response 28: Thank you for your comments. We have revised it.
Line 290: Oxford comma missing before “and”
Response 29: Thank you for your comments. But we do not find the missing Oxford comma in Line 290.
Line 367: Species name not italicized
Response 30: Thank you for your comments. We have revised it.
Line 379: Oxford comma missing before “and”.
Response 31: Thank you for your comments. We have revised it.
Line 393-405: I am not pleased with this explanation (specially with using an example of richness to support a difference found in diversity). Authors found a pattern that contradicts what is known from literature. While species differences might be the reason for it, it is too simple of an explanation. I would like to know whether the level of disturbance between these studies is equivalent. For instance, are the disturbed areas in the previous studies agriculture related (with a lot of pesticides that kill a lot of micro-organisms), or factory pollution (also kills a lot of micro-organisms)? Also, humans (tourists) carry a lot of different bacteria with them, so in this scenario it would not surprise me that a different source of disturbance would lead to contradictory results in terms of diversity.
Response 32: Thank you for your comments. After careful consideration, we felt that this paragraph is repeated with the following 4.4, so we decided to delete this paragraph and merge it into 4.3 after modification.
Line 391-414: This paragraph has some repetitions (for instance Bd) and back and forwards. I would recommend streamlining the information some more.
Response 33: Thank you for your comments. We have revised it.
Line 423-424: Please clarify that it was water temperature
Response 34: Thank you for your comments. The expression here is as the water temperature and the temperature rise. So we revised it to the water temperature and air temperature.
Line 427: Species name not italicized
Response 35: Thank you for your comments. We have revised it.
References:
Line 563-565: Do not capitalize every word of the title (confirm this in all references as it happens quite frequently), specially not the specific epithet.
Response 36: Thank you for your comments. We have revised them.
Line 721-722: Genus name not italicized
Response 37: Thank you for your comments. We have revised it.
Line 727-729: Please remove the hyperlink
Response 38: Thank you for your comments. We have revised them.
Figure 2: I would suggest adding a label to the legends in the figure saying “Phyla” and “Families” for clarity.
Response 39: Thank you for your comments. We have revised it.
Supplementary Tables: Species name is not italicized in the title.
Response 40: Thank you for your comments. We have revised it.
Comments on the Quality of English Language
While the manuscript is well written in general, some sections (see line-by-line comments above) are repetitive and/or unclear. The manuscript needs some minor editing when it comes to grammar.
Response 41: Thank you for your comments. We have revised it line-by-line.